# Rare disease in Malaysia: Challenges and solutions

**Asrul Akmal Shafie**[1]*, **Azuwana Supian**[1,2], **Mohamed Azmi Ahmad Hassali**[1], **Lock-Hock Ngu**[3], **Meow-Keong Thong**[4], **Hatijah Ayob**[5], **Nathorn Chaiyakunapruk**[6]

1 Discipline of Social and Administrative Pharmacy, School of Pharmaceutical Sciences, Universiti Sains Malaysia, Minden, Penang, Malaysia, 2 Pharmaceutical Services Division, Ministry of Health Malaysia, Petaling Jaya, Selangor, Malaysia, 3 Genetic Clinic, Hospital Kuala Lumpur, Kuala Lumpur, Malaysia, 4 Genetics and Metabolism Unit, Department of Paediatrics, Faculty of Medicine, University of Malaya, Kuala Lumpur, Malaysia, 5 Malaysian Rare Disorders Society, Petaling Jaya, Selangor, Malaysia, 6 Department of Pharmacotherapy, College of Pharmacy, University of Utah, Salt Lake City, Utah, United States of America

* aakmal@usm.my

**Data Availability Statement:** All data are available in the manuscript.

**Funding:** The authors received no specific funding for this work.

## Abstract

### Objective

Rare diseases are often underdiagnosed, and their management is frequently complicated by a lack of access to treatment and information about the diseases. To allow for better policy planning, we sought to examine the current status of managing rare diseases in Malaysia.

### Methods

This study was conducted in two phases. In the first phase, we triangulated information from reviews of journal publications, documents from the Malaysian government and in-depth interviews among selected key healthcare stakeholders in Malaysia. The second phase was designed as a cross-sectional survey to estimate the number of cases and treatment coverage for rare diseases in Malaysia.

### Results

Malaysia has no official definition of rare disease yet but currently in the process of reviewing them for Malaysia. There are 13 rare disease specialists and a dozen medical doctors in genetic clinics around Malaysia, mainly in public health facilities. From the survey, 1,249 patients were diagnosed with rare diseases in public hospitals. Only 60% received their medications or supplements, and the rest continued with symptomatic treatment.

### Conclusion

Generally, Malaysia has made significant progress in the management of rare diseases, but there are still opportunities for development in critical areas. Ultimately, if all healthcare providers, government, society, and politicians work together to manage rare diseases, we will see an improvement in patient outcomes.

**Competing interests:** The authors have declared that no competing interests exist.

**Abbreviations:** DRGD, Drug Registration Guidance Document; ERT, Enzyme replacement therapy; GDP, Good distribution practice; GMP, Good manufacturing practice; HKL, Hospital Kuala Lumpur; HUSM, Hospital Universiti Sains Malaysia; ICAP, International Compassionate Assistance Program; ICD, International Classification of Diseases; IEM, Inborn errors of metabolism; IMR, Institute for Medical Research; MLDA, Malaysia Lysosomal Diseases Association; MMS, Malaysia Metabolic Society; MNMP, The Malaysian National Medicines Policy; MOH, Ministry of Health, Malaysia; MRDS, Malaysian Rare Disorders Society; MYR, Malaysian ringgit; NGO, Non-governmental organisation; NPRAM, National Pharmaceutical Regulatory Agency Malaysia; SJMC, Subang Jaya Medical Centre; SMA, Spinal muscular atrophy; THB, Thai baht; UHC, Universal health coverage; UKMMC, Universiti Kebangsaan Malaysia Medical Centre; UMMC, University Malaya Medical Centre; WHO, World Health Organisation.

## Introduction

The term "rare disease" is used to identify any illness that affects a small percentage of the population and is frequently debilitating [1]. There are approximately 6,000 to 8,000 rare diseases worldwide. Most of the illnesses are associated with genetic disorders and remain undertreated because many healthcare professionals do not recognise the symptoms or, in some instances, patients are not aware of the proper channels for treatment [2]. Research has shown that some rare diseases have a more harmful effect on health-related quality of life (QoL) than other serious illnesses. Many times, the patients' psychosocial abilities are impaired despite appearing to be physically healthy [3]. Rare disease patients are characterised by the uniqueness of their illness (less than 1:2,000 in Europe), as rare diseases collectively affect 1 in every 15 persons or 400 million people worldwide [4–6].

Rare diseases are difficult to diagnose due to their heterogeneous and variable presentations as well as a lack of local population data on the conditions of the diseases [7]. In addition, a lack of expertise and laboratory support services often results in misdiagnosis as well as delayed diagnosis of rare conditions [8]. The lack of awareness by healthcare providers and limited resources are some of the significant challenges facing rare disease patients as well as their families [9]. Moreover, the lack of awareness and scarcity of resources leads to a shortage of trained medical professionals in the area of rare diseases. Due to the lack of exposure, many healthcare providers believe they are unlikely to come across rare disease cases in their professional careers [9]. Hence, rare disease patients often lack access to adequate treatment compared to patients with other chronic illnesses [10].

Another problem with the diagnosis of rare diseases is the absence of proper coding that is specific to rare diseases. The current International Classification of Diseases (ICD) system that is used in most countries does not have the same classification and coding for new rare diseases [2, 11]. There are only approximately 500 codes that are specific to rare diseases in the ICD version 10 (ICD-10) [12, 13]. As a result, standard coding and registry for rare disease cases cannot be carried out throughout different countries. The rare disease registry is essential, as it would permit regulatory authorities to track, follow up, and evaluate a disease quickly [7, 14]. The Commission of European Communities urges the establishment of registries and databases of rare diseases, as they will be the primary mechanisms to increase knowledge on and management of rare diseases [15]. Recently, the WHO released the ICD version 11 (ICD-11) in June 2018, which listed 5,400 unique codes for rare diseases [16]. Orphanet, which uses the European definition of rare disease, also provides a unique and stable identifier called the ORPHA number to classify rare diseases [17]. The adoption of both types of international codes (ICD-11 and ORPHA number) for the registry of rare diseases could improve the country's record-keeping and monitoring of data.

Given the current environment and its complexities, there is a need to understand rare diseases in Malaysia, as they are seldom discussed among policymakers and the general public. The discussion is often hampered by a lack or total absence of information on the epidemiology and burden of rare diseases. To achieve equitable access to quality, safe, effective and affordable medicine, as outlined in the Malaysian National Medicines Policy, Malaysia must take rare disease issues more seriously and offer suggestions on how to tackle rare diseases better, thus potentially improving healthcare [18]. Therefore, this study aims to determine the current healthcare status and baseline information about rare diseases in Malaysia.

## Methodology

### Ethics approval and consent to participate

The study was approved by the following:

1. Medical Research & Ethics Committee, Ministry of Health Malaysia (NMRR-14-1880-22900 (IIR).

2. Director of Kuala Lumpur Hospital, Malaysia (CRCHKL Reg. No: HCRC.IIR-2016-04-071.

All participants provided written informed consent before entering the study.

The study was conducted using mixed methods, which were divided into two phases. In the first phase, we used multiple sources to obtain relevant information related to rare diseases in Malaysia, including communication with the Ministry of Health Malaysia (MOH), a literature review and in-depth interviews. The literature review was conducted by mainly using PubMed, Google Scholar, the Malaysian government's documents and rare disease society websites. Meanwhile, in-depth interviews were conducted with selected healthcare stakeholders.

For stakeholder interviews, 29 key opinion leaders were identified through the government's channels and expert connections. This group included policymakers, geneticists (including hospital pharmacists), advocacy groups, and industry leaders who are involved in rare disease and orphan drug management. The semi-structured interview focused on two main issues (Table 1): i) issues and problems surrounding rare diseases in Malaysia and ii) issues and challenges regarding orphan drugs in Malaysia. The interviews were conducted face-to-face from April to August 2016 using a voice recorder and interview guidance and took approximately 40 to 90 minutes for each respondent.

In the second phase, a list of rare diseases based on the World Health Organisation (WHO), which consists of 28 groups as well as 351 types of rare diseases, was used in a cross-sectional survey to estimate the access to treatment for rare disease patients in Malaysia. The study was distributed to ten major tertiary hospitals from both the MOH and university teaching hospitals. These hospitals are key references in Malaysia, where specialists and consultants in paediatrics and genetics (i.e., paediatricians and geneticists) are located. The list of rare diseases was tabulated in a simple format according to five segments: a) group of disorders, b) rare diseases, c) number of patients, d) number of patients treated and f) treatment involved for the rare diseases. This feedback allowed more precise baseline data for rare diseases and their treatments and current orphan drug access in Malaysia. Because rare diseases are rare and difficult to document, the survey was conducted from July 2014 until June 2015.

**Table 1. Characteristics of the respondents and content of the topic guide in an interview session.**

| Respondent | Number of Respondents | Topic guide |
|---|---|---|
| Policymakers (PM) | 7 | • Definition of rare disease and orphan drug<br>• Policy and guideline of rare disease and orphan drug<br>• Orphan drugs in the National Formulary |
| Genetic Clinic (GC) | 7 | • Status of rare disease in Malaysia<br>• Awareness among physicians and patients<br>• Challenges for rare disease and orphan drug |
| Advocacy groups (AG) | 3 | • Issues and problems of rare disease management<br>• Challenges in societies and advocacy groups |
| Pharmaceutical industry (PI) | 12 | • Registration of orphan drug<br>• Issues and challenges of orphan drug management |

### Analysis

We adapted the themes in the WHO's Framework for Action in Strengthening Health Systems [19] to triangulate the data from the literature review, in-depth interviews and cross-sectional surveys. All information was extracted based on themes that were modified from this framework. It consisted of five themes that were used to classify and evaluate the national initiatives of rare diseases in Malaysia: a) healthcare system and governance; b) rare disease awareness and patient advocacy; c) clinical expertise and patient management; d) funding; and e) newborn screening of rare diseases. The audio recordings from the in-depth interviews were transcribed verbatim. ATLAS. TI (version 8) was used to conduct a thematic analysis on participant transcripts and facilitate the handling of the data. Data from the cross-sectional survey were analysed using Microsoft Excel.

## Results

The results from both phases were summarised according to the following five themes: a) healthcare system and governance; b) clinical expertise and patient management; c) newborn screening; d) funding; and e) rare disease awareness and patient advocacy. A total of six hospitals from 10 facilities sent their feedback on rare diseases and their treatments. Three hospitals were from MOH facilities: Hospital Kuala Lumpur (HKL), Hospital Raja Permaisuri Bainun and Hospital Sultanah Nur Zahirah. The other three hospitals were from university hospitals: University Malaya Medical Centre (UMMC), Universiti Kebangsaan Malaysia Medical Centre (UKMMC) and Hospital Universiti Sains Malaysia (HUSM).

A total of twenty-six respondents were successfully interviewed in this study. Seven respondents were policymakers (three officers from the Pharmacy Policy & Strategic Planning Division, two officers from the Pharmacy Practice & Development Division and two officers from the National Pharmaceutical Regulatory Agency). Another four respondents were from genetic clinics (three from the MOH and one from a university hospital), and 12 respondents were from the pharmaceutical industry as licensed holders of orphan drugs. Meanwhile, three respondents from the advocacy group were the presidents of the three leading rare disease societies in Malaysia (Malaysian Rare Disorders Society, Malaysia Lysosomal Diseases Association and Malaysia Metabolic Society). However, two respondents from genetic clinics did not respond, and one was on an extended leave. Table 1 describes the characteristics of the respondents.

### Malaysian healthcare system and governance

An effective healthcare system is considered a central factor for the successful development of a country. Malaysia is one of the few developing countries that has implemented universal health coverage (UHC). With a total estimated population of 31.6 million in 2016, MYR 23 billion was spent on health, which is approximately 8.6% of the national budget (Table 2).

To date, there is no official definition of rare disease in Malaysia, even though the most prominent rare disease voluntary organisation called the Malaysian Rare Disorders Society (MRDS) defines rare disease as a disease that affects less than 1 in 4,000 people in a community [20]. However, upon request, the MOH provides us with a definition that they have internally adopted i.e. *"any disease that is prevalent in 0.65% to 1% of the population, and the diagnosis and treatment are complicated"* purportedly from the World Health Organisation (WHO). Despite the reference, we are unable to trace the WHO documents and publications that could support the definition. Currently, the MOH still in the process of reviewing the rare disease definition in Malaysia.

**Table 2. Malaysian healthcare system 2012–2016.**

| | Indicators | 2012 | 2013 | 2014 | 2015 | 2016 |
|---|---|---|---|---|---|---|
| Socioeconomic | Per capita GDP, MYR | 25,345 | 26,473 | 27,484 | 37,123 | 38,887 |
| | Health budget (million), MYR | 16,871 | 19,277 | 21,993 | 23,312 | 23,031 |
| | Percentage of health budget allocation to national budget (%) | 7.25 | 7.66 | 8.33 | 8.51 | 8.62 |
| | Estimated population (million) | 29.34 | 29.71 | 30.71 | 31.19 | 31.66 |
| Health Services | Number of public hospitals (MOH & non-MOH) | 147 | 149 | 150 | 152 | 153 |
| | Number of doctors (public) | 27,478 | 35,219 | 33,275 | 33,545 | 36,403 |
| | Number of doctors (private) | 11,240 | 11,697 | 12,290 | 12,946 | 13,684 |
| | Ratio of doctor to population | 1: 758 | 1: 633 | 1: 661 | 1:656 | 1:632 |

Source: MOH and Department of Statistics Malaysia

"*The definition of rare disease is very important. . . a key point to start a discussion on rare disease management. Unfortunately, we do not have a fixed definition for rare disease. It is difficult to define because we don't have the statistics or prevalence data. So, we use and follow the definition of rare disease from WHO for now.*"

*(PM-A7)*

The second edition of the Malaysian National Medicines Policy (MNMP 2012) defines an orphan drug as a medicine, vaccine or in vivo diagnostic agent that is intended to treat, prevent or diagnose a rare disease. An orphan drug is identified as medicine that is not commercially viable to supply, treat, prevent or diagnose another disease or condition [18].

One of the oldest legislation in Malaysia is the Sales of Drugs Act 1952. It consists of the Control of Drugs and Cosmetics Regulations 1984, which states that all products (drug) need to be registered to sell and supply the product in Malaysia [21]. Orphan drugs also have to fulfil all the requirements of good manufacturing practice (GMP) and good distribution practice (GDP) to ensure the quality, safety and efficacy of products.

"*. . . orphan drugs also have to follow the major components of GMP and GDP. It must comply with the quality, safety, and efficacy requirements. However, there is flexibility in the number of documents to submit for registration. . . upon discussion with NPRA and on a case-to-case basis.*"

*(PM-A1)*

In September 2016, the National Pharmaceutical Regulatory Agency (NPRA) updated its Drug Registration Guidance Document (DRGD). This second edition, which served as the reference guide for the registration process, includes a new section that is specific to the registration of orphan products [22]. Based on this guidance, an orphan product is defined as in the MNMP 2012 and registered with a unique code or a particular alphabet for surveillance and monitoring. For example, the code MAL———AY is for poisons, and MAL———XY is for supplements.

The most vital element for rare disease management is effective governance to execute a proper national strategy. Patient registries are critical in disease management. However, there is no national rare disease registry in Malaysia; each centre keeps its patient registry. Hence, there is no information on prevalence of rare diseases in Malaysia.

"*I agree that the Malaysian Commission should propose legislation to regulate rare diseases. Currently, such a registry is not available. It is important to provide information on the prevalence and incidence rate of a specific disease to make the right decision on disease management.*"

*(PI-C3)*

"*. . . it is not easy to do a prevalence study. It requires many clinical experts in rare diseases and costs a lot of money. It is difficult to diagnose, especially ultra-rare disease. The laboratory tests are also not easily accessible, and some have to be sent abroad.*"

*(GC-B2)*

The results from the cross-sectional survey show that 1,249 patients were diagnosed with rare diseases, and approximately 60% received their treatment based on the WHO list of rare diseases (Fig 1). Out of the 28 groups of rare diseases, congenital syndromes, disorders of energy metabolism and lysosomal disorders are the top three rare disease groups. Each group has more than one hundred patients.

1. Congenital syndromes, 2. Disorders of energy metabolism, 3. Lysosomal disorders, 4. Urea cycle disorders and inherited hyperammonaemia, 5. Disorders affecting bone, cartilage and connective tissue, 6. Neurological disorders, 7. Disorders of the metabolism of amino acids not classified as organic acidurias, 8. Organic acidurias, 9. Muscle disorders, 10. Disorders of the metabolism of vitamins and (non-protein) cofactors, 11. Disorders of carbohydrate metabolism, 12. Disorders of neurotransmitter metabolism, 13. Haematological and immunological disorders, 14. Peroxisomal disorders, 15. Disorders of the metabolism of purines, pyrimidines and nucleotides, 16. Pulmonary disorders, 17. Gastrointestinal and hepatological disorders, 18. Disorders of the metabolism of trace elements and metals, 19. Endocrine disorders, 20. Dermatological disorders, 21. Renal disorders, 22. Disorders of porphyrin and haem metabolism, 23. Disorders of fatty acid and ketone body metabolism, 24. Disorders of lipid and lipoprotein metabolism, 25. Disorders of the metabolism of sterols, 26. Disorders of amino acid transport, 27. Congenital disorders of glycosylation and other disorders of protein modification and 28. Other disorders of amino acid and protein metabolism.

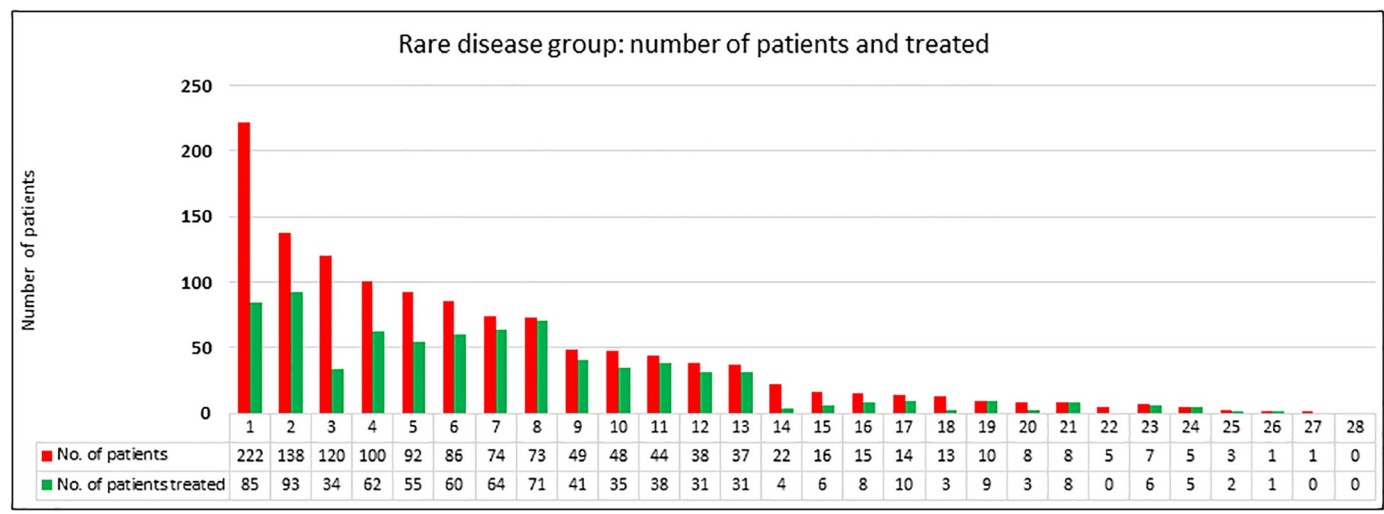

| | 1 | 2 | 3 | 4 | 5 | 6 | 7 | 8 | 9 | 10 | 11 | 12 | 13 | 14 | 15 | 16 | 17 | 18 | 19 | 20 | 21 | 22 | 23 | 24 | 25 | 26 | 27 | 28 |
|---|---|---|---|---|---|---|---|---|---|----|----|----|----|----|----|----|----|----|----|----|----|----|----|----|----|----|----|----|
| No. of patients | 222 | 138 | 120 | 100 | 92 | 86 | 74 | 73 | 49 | 48 | 44 | 38 | 37 | 22 | 16 | 15 | 14 | 13 | 10 | 8 | 8 | 5 | 7 | 5 | 3 | 1 | 1 | 0 |
| No. of patients treated | 85 | 93 | 34 | 62 | 55 | 60 | 64 | 71 | 41 | 35 | 38 | 31 | 31 | 4 | 6 | 8 | 10 | 3 | 9 | 3 | 8 | 0 | 6 | 5 | 2 | 1 | 0 | 0 |

**Fig 1. Rare disease groups: Number of rare disease patients and number of rare disease patients treated in Malaysia.**

**Table 3. Top 10 rare diseases in Malaysia, number of patients, number of patients treated and its treatments.**

| Rare diseases | No. of patients | No. of patient treated | Treatments |
|---|---|---|---|
| Marfan syndrome | 81 | 41 | Losartan, oral propranolol, supportive care |
| Prader-Willi syndrome | 60 | 10 | Growth hormone therapy, supportive care |
| OI | 45 | 44 | Sodium pamidronate |
| MELAS | 39 | 23 | Oral L-carnitine, Coenzyme Q10, Arginine |
| MPS type II | 39 | 9 | Idursulfase (ERT) |
| Citrullinemia type II | 33 | 3 | Arginine, Sodium pyruvate, diet |
| MSUD | 33 | 33 | Sodium phenylbutyrate |
| Leigh syndrome | 32 | 22 | Coenzyme Q10, Riboflavin, Thiamine |
| GSD type Ia | 31 | 26 | Allopurinol |
| DMD | 26 | 23 | Prednisolone |

OI (Osteogenesis imperfecta), MELAS (Mitochondrial encephalomyopathy lactic acidosis and stroke-like episodes), Mucopolysaccharidosis type II (MPS II), MSUD (Maple Syrup Urine Disease), GSD (Glycogen Storage Disease), DMD (Duchenne Muscular Dystrophy)

In the congenital syndromes group, Marfan syndrome had the highest number of patients (81 patients), followed by Prader-Willi syndrome (60 patients) and Noonan syndrome (16 patients). Mitochondrial encephalomyopathy lactic acidosis and stroke-like episodes (MELAS) had the highest number of patients (39 patients) in the disorders of energy metabolism group, followed by Leigh syndrome caused by mutations in nuclear genes (32 patients) and myoclonic epilepsy associated with ragged-red fibres (MERRF) (12 patients). Meanwhile, in the lysosomal disorders group, Hunter syndrome or Mucopolysaccharidosis type II (MPS II) was the highest (39 patients), Mucopolysaccharidosis type IVA (MPS IVA) (19 patients) was the second-highest, and Pompe disease or Glycogen storage disease type II (GSD II) (17 patients) was the third-highest.

However, not all rare disease patients are receiving treatment. Only 85 of 222 patients were treated in the congenital syndromes group (38.3%), 93 of 138 patients were treated in the energy metabolism group (67.4%), and only 34 of 120 patients were treated in the lysosomal disorders group (28.3%). Table 3 shows that Marfan syndrome, Prader-Willi syndrome, and Osteogenesis imperfecta are the top three rare diseases in Malaysia.

## Clinical expertise and patient management

The majority of health facilities in Malaysia do not offer genetic clinic (rare disease) services. Only four hospitals offer genetic and metabolic clinic services: one from the MOH and another three from university hospitals. Generally, the awareness level among Malaysian physicians about rare diseases is still low compared to their familiarity with other conditions. This is mainly due to the small degree of rare disease cases in their clinics or hospitals.

*". . .very few physicians know about rare diseases. They might have seen or handled some Marfan syndrome (rare disease) cases because they are easy to diagnose. But rare diseases other than that are not very familiar and not easy to diagnose."*

*(GC-B2)*

*"Most physicians are not aware. . . of either rare diseases or orphan drugs because of no available data. It only started in UMMC in the year 2004 when we created awareness among healthcare providers such as physicians, general practitioners, pharmacists, nurses and others."*

*(GC-B1)*

In Malaysia, there is a limited number of genetic specialists and only a few hospitals that offer specific services to treat rare diseases. The first dedicated genetic service with genetic counselling was at UMMC in 1995. The subspecialty of "Clinical Genetics" was recognised in the National Specialist Register [8]. Overall, Malaysia has only 13 rare disease specialists, with a dozen medical doctors in five health centres. Twelve are located in public hospitals, eight in HKL from MOH and the other four in the Ministry of Education: UMMC (1), UKMMC (1) and HUSM (2). Meanwhile, there is only one geneticist practising in a private hospital at Subang Jaya Medical Centre.

HKL, which is the primary reference hospital in Malaysia, has the highest number of genetic and metabolic disease specialists and medical officers. The MOH takes the initiative to expand their services by appointing two senior specialists who have been appointed to visit once a month or two to six months to other selected MOH hospitals by region in Malaysia. All of these specialists have contributed significantly to the country and society, especially to rare disease patients.

*". . . for patients in the Ministry of Health, especially those who have received enzyme replacement therapy (ERT), all of them are centred in Hospital Kuala Lumpur. We have the expertise. . . geneticist consultants, specialists, special nurses and dietitians here. However, we do visit the other six hospitals (MOH facilities) in four states. . . Johore, Penang, Sabah and Sarawak."*

*(GC-B2)*

Genetic and inherited metabolic disorders are included in the syllabus of a five-year medical degree programme at local public universities. Furthermore, the training of the first two qualified genetic counsellors started recently at UMMC, in collaboration with Australian universities. The Master of Science programme in genetic counselling has been offered locally since 2015 at UKMMC.

*"The medical degree students have learned genetics in detail, and rare disease was used as an example of the application of genetics in medicine. Recently, we also offer an advanced course for counsellors, specifically in genetic counselling."*

*(GC-B1)*

*"We need genetic counsellors. They play a big role in genetic counselling to the patients and families, especially when they are diagnosed with rare diseases. This session will explain what the disease is all about, how it has been inherited, any treatment options, disease prognosis, and option for future pregnancy and encourage early screening."*

*(GC-B2)*

In Malaysia, rare disease awareness is still lacking among nurses. Nurses should be trained and well informed to identify early signs and symptoms of rare diseases. They are the frontliners for routine schedule check-up, vaccinations, and anthropometric intake (weight and height) to monitor the growth and developmental assessment of children. Services are generally administered and handled by nurses in public health clinics until babies reach the age of six. With their knowledge, any unusual sign and symptom or problem that requires further clarification can be referred to the physicians.

"*It should start in the public health clinic. . . right from the beginning. . . when they go for routine monthly check-ups. The growth (height) of some of these children is not the same as normal children, so it should be done in public health clinics before coming to the hospitals.*"

*(AG-D2)*

## Newborn screening

Newborn screening began in 1980 with cord-blood screening for glucose-6-phosphate dehydrogenase deficiency [23]. An expanded newborn screening programme includes testing for inherited metabolic diseases involving amino acid metabolism, fatty acid oxidation, and organic acid metabolism disorders. However, it is not compulsory for all newborn babies. A lack of awareness among clinicians and laboratory diagnostic facilities has resulted in under-reported cases. Since the establishment of diagnostic facilities in 1999, Malaysia has expanded newborn screening to include inborn errors of metabolism.

Nevertheless, the expanded tests are only available in four centres, which are the Institute for Medical Research (IMR), Hospital Kuala Lumpur (HKL), UMMC and the Centre for Advanced Analytical Toxicology Services (research and service laboratory in Universiti Sains Malaysia) on request by the public and some private hospitals [23]. An epidemiology study published in 2008 on the selected testing of ill infants and children for IEM yielded a positive result of 2% (265/13,500) for IEMs in Malaysia [24]. Another pilot study at the Institute for Medical Research (IMR) published in 2016 showed that the detection rate of inborn errors of metabolism (IEM) was 1 in 2,916 newborns [25]. Although the figure does not reflect the actual total incidence of IEM in Malaysia, it is comparable to those in other Asian countries, such as Singapore (1 in 3,165) and South Korea (1 in 2,800) [26, 27]. Nevertheless, many of the rare disease tests had to be sent abroad to Australia, Japan and Taiwan for analysis, which adds to the total cost of treatment.

"*Majority of genetic syndromes need molecular diagnostics where the test is not accessible in Malaysia. It can be sent overseas, but it is expensive (to do so). HKL and IMR are mainly for inborn errors of metabolism. However, there are thousands of different genetics. . . we only have few available here. Some of the tests still have to be sent abroad, like to Australia and Taiwan.*"

*(GC-B2)*

Genetic screening is offered to the siblings of confirmed rare disease patients and provided to other relatives. Female sibling screening will be performed when she reaches the age of 17 or after she finishes secondary school. However, the results are only given to the parents since they have the custody to report the effects and consequences to their daughter. In this situation, a genetic counsellor can play a role in counselling parents and their children.

## Funding

The major challenge in rare disease management is financial limitation. Unfortunately, due to the rare nature of the diseases, they receive limited attention from funders. As a result, the government allocates only a minimal amount of funds for managing rare diseases.

"*Funding from the government for rare disease management is not enough. From the current economic climate, there is nothing much that we can change.*"

(GC-B1)

"*The special allocation for the genetic clinic in HKL only started in 2007. We are managing, except for very expensive orphan drugs. Some of them need ERT summarised, which costs more than one million Malaysian ringgit. Unfortunately, the budget is definitely not enough for all patients. We always advocate for the benefit of our patients. Yes. . . it is expensive, but I don't think it is inaccessible. . . it is just not a priority from the authorities.*"

(GC-B2)

Currently, all of the government's budgets for rare diseases are centred in HKL, the reference hospital for MOH. Fig 2 shows the allocation received by the Genetic Clinic in HKL from 2013 to 2016. This allocation is only for medicines that are specific to the treatment of rare diseases and does not include other standard medicines or items. Although the allocations are increasing by 7% to 10% each year, they are not sufficient to cover the high number of rare disease patients. Furthermore, there are no individual allocations for rare diseases in other university hospitals.

"*All rare disease patients in university hospitals who need the treatment of ERT have to be transferred to HKL. To me, it should not be that way. University hospitals should also be given the opportunity to treat these patients. They are also trained doctors, and as teaching hospitals, they can do some research on the disease.*"

(AG-D2)

Some of the rare disease patients need more than half a million ringgit every year for their treatment costs. As an example, *idursulfase* (ERT) for the treatment of Mucopolysaccharidosis type II (MPS II) costs approximately MYR 6,800 per vial. A patient who weighs 25 kg would need two vials per week, which requires more than MYR 700,000 every year. Some of the other orphan drugs have used up to MYR 1.6 million in funds for only one ERT each year. In addition to government funding, the patients have also received the ERT from other parties, such as pharmaceutical industries, other corporate bodies and non-government organisations

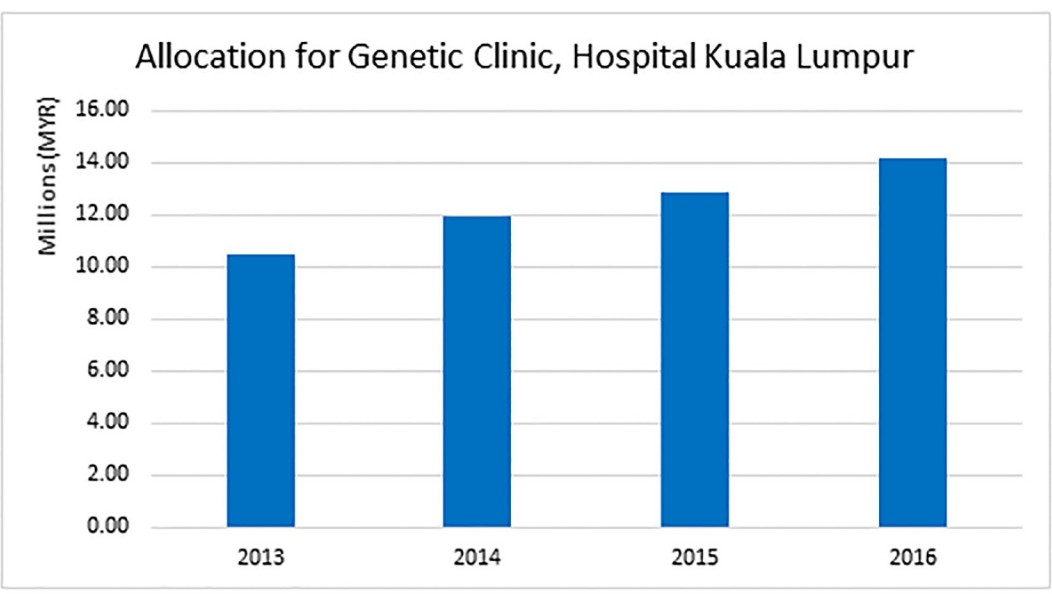

**Fig 2. Allocation for genetic clinic in Hospital Kuala Lumpur (2013–2016).**

(NGOs), but it is far from enough. The fund is only secured for a 6-month term and hence lacks sustainability.

> "*Some of our patients received ERT from the International Compassionate Assistance Programme (ICAP) from pharmaceutical industries that manufacture and market the ERT. A few of our patients also received it from NGOs and other local corporate companies. However, this is just temporary and can end anytime. No country in the world gives this responsibility to other parties. It is the government's accountability.*"
>
> (GC-B2)

> "*No. . . we cannot pay for the treatment. How on earth can patients afford to pay this kind of money? It costs millions of ringgit. . . every year. . . for the rest of their lives. I heard that the budget cap is the same amount since 2010. These children are reaching puberty, their age, weight. . . they keep growing, and they need higher doses. But the situation now seems that they can only get the same doses as those five years back. . . definitely under dosed.*"
>
> (AG-D1)

> "*This one patient. . . she couldn't get the drug allocation from the government. So, as a rare disease advocacy group, we want to change history. The Malaysia Lysosomal Diseases Association (MLDA) raised the funds and paid 50% of her drug cost . . .it was about MYR 816,000. Another 50% was from the industry.*"
>
> *(AG-D1)*

> "*The government fund is beneficial but not enough. As rare diseases normally affect a small segment of the population, the government should have the capacity to allocate special funding to ensure that this group of patients has access to optimal treatment.*"
>
> *(PI–C3)*

Nevertheless, not all rare diseases require expensive treatments. Table 2 shows that some rare diseases only need standard medicines such as *losartan*, *prednisolone* and *allopurinol*; supplements, such as *thiamine*, *riboflavin*, *carnitine* and *coenzyme Q-10*; and special milk or diets.

> "*Rare disease patients. . . they are getting very good treatments from the MOH. If you talk about treatments like conventional medicines and inexpensive and special milk, they are all still provided by the government, including supplements such as coenzyme Q10. There are no problems except with ERT because the costs are too large, so our system still cannot cater (to it).*"
>
> (AG-D1)

## Rare disease awareness and patient advocacy

The awareness of rare diseases in Malaysia is relatively low but has slowly gained momentum over the last ten years. It is a critical factor in rare disease management, as it leads to greater public and private involvement. Usually, rare disease awareness campaigns are driven by patient advocacy groups and NGOs. Patient advocacy from both the public and private sectors plays an essential role in gaining attention and involvement from the government or policymakers.

"*Everywhere we go, people are not aware of rare diseases, including the doctors. I think they know about rare diseases because they learn about them in medical school but only because of that, since they have not seen the cases and are not aware if there is a rare disease case. So, I took the challenge to form a rare disease support group and have been networking with parents since 2004. Then, we became a registered society in 2007 and started a few activities and projects. We learned a lot from The Taiwan Foundation for Rare Disorders.*"

*(AG-D2)*

In Malaysia, there are three foremost rare disease advocates: Malaysian Rare Disorders Society (MRDS), Malaysia Lysosomal Diseases Association (MLDA) and Malaysia Metabolic Society (MMS). Additionally, there are many small societies under MRDS, such as the Welfare Association for Dwarfism Malaysia, Spinal Muscular Atrophy (SMA) Malaysia, We Care Journey and others. They host awareness campaigns and local rare disease conferences, produce information booklets, and join media broadcasts such as television programmes, publications, and blogs as well as social media. Every year they organise rare disease days, charity runs, fundraising and donations.

"*MLDA started in 2011. We have raised funding on our own. . . charity dinners and a few campaigns to get more public support. For the last five years, only 17 lysosomal patients were treated with ERT, and now there are already 30 patients . . . I think that is the major achievement for our society.*"

*(AG-D1)*

"*All the costs for treatment should be from the government's budget. We as support groups and societies have to think about and handle these social issues. Not only the patients but also the caregivers. Now, we are actively raising funds for genetic testing, especially for the patients' family members.*"

*(AG-D2)*

In 2011, MRDS organised the first rare disease conference in Malaysia. The meeting was co- organised with the Medical Genetics Society of Malaysia. It was followed by the second conference two years later. In the meantime, MLDA officially launched a long-term campaign, "RM1 for Every Life Counts" in May 2016. The primary purpose of the campaign was to support and sustain the ERT for current patients, as well as to sponsor patients who are still waiting for their treatment [28].

Malaysian rare disease advocacy groups also joined numerous collaborations within the Asia Pacific region. Rainbow Across Borders became the coordinator and is based in Singapore since 2015 [29]. This organisation promotes regional collaboration and networking among patient support organisations and focuses on developing the rare disease registry and directory across Asia.

"*All those rare disease registries need to be consolidated; then you have stronger databases. I have attended the Asia Conference for rare diseases led by Rainbow Across Borders. We are just a small society. We should collaborate with other Southeast Asian countries. If all health ministries can compile the ERT needed, then we will have the volume. Maybe we can bring down the drug prices.*"

*(AG-D1)*

## Discussion

In the management of rare diseases, the definition is fundamental to kick off further actions. Each country has its definition of rare diseases according to its prevalence, and it varies across regions. For example, in the USA, a disease is rare when it affects fewer than 200,000 people [30]. In the European Union, a disease is considered a rare disease when it affects less than one in 2,000 people [31]. Whereas in East Asia, Taiwan defines a rare disease if it occurs fewer than one in 10,000 population [11]. The majority of the countries defined rare diseases as to be between 0.01% to 0.05% of their population. These percentages are far below the one used by the MOH (0.65% to 1%). However, there are doubts on the definition used by the MOH given that there is no reference that could support it—possibly a 'folklore' definition. The definition of rare disease cannot be arbitrary as it could have a wrong impression, create misunderstanding and complicate discussion among rare disease stakeholders, especially the patients and caregivers. Nevertheless, the MOH now reviewing the country's rare disease definition to be in line with other reference countries as well as neighbouring countries, which will encourage cooperation in matters related to the management of rare diseases in the future.

In 2016, Helen Clark, the former administrator of the United Nations Development Programme stated at the 11th Annual International Conferences on Rare Diseases and Orphan Drugs that, "No country can claim to have achieved UHC if it has not adequately and equitably met the needs of those with rare diseases" [32]. This statement is very relevant to countries that implement total universal health coverage, such as Malaysia. The basic principle of UHC is that all diseases should be treated with readily available and essential treatments. Teerawattananon et al. (2016) suggested four building blocks for setting the effective priority of UHC [33]:

1. Governing structure with precise functions and regulation of institutes and their inter-relation;

2. Capacity building programmes by policymakers, researchers, and other stakeholders, including the general public for a better understanding of health priority;

3. Resource availability and mobilisation to support priority setting; and

4. Collaboration with networks of local, international, and global organisations that aim to support UHC policies.

These building blocks can help the country determine feasible requirements, such as services, resources, and financial means, to achieve total UHC.

### Precise functions and regulation of governance

A disease registry is significant in disease management as a tool for tracking patients' clinical care, outcomes, safety, and comparative effectiveness. It is used to store individual personal information and medical history for health policy purposes [34]. In Malaysia, there are approximately 20 national disease registries, such as the Diabetes Registry and Cancer Registry. Unfortunately, there is still no national registry for rare diseases. One of the factors for its absence is the limited codes available for a rare disease in ICD version 10 (ICD-10), which was implemented in Malaysia. There are only approximately 500 codes specific to rare disease in ICD-10 [12, 13]. Recently, the WHO released ICD version 11 (ICD-11) in June 2018, which lists 5,400 unique codes for rare disease [16]. In addition, Orphanet, which uses the European definition of rare disease, also provides a unique and stable identifier called the ORPHA number to classify rare disease [17]. The adoption of both international codes (ICD-11 and

ORPHA number) for a rare disease registry could improve the country's record-keeping and monitoring of data.

The figures reported in this study might not be generalisable to all of Malaysia because the data was only collected from public health facilities (MOH and university hospitals). To obtain official prevalence data for the country, a population study needs to be included for every rare disease for a prolonged period with funding and collaboration from all healthcare providers in the public and private sectors.

## The capacity of all health stakeholders to build programmes

Rare disease patients usually face a few challenges, such as a lack of specialists in rare diseases and a lack of knowledge among healthcare providers [35]. There is a shortage of geneticists, metabolic disorder specialists and genetic counsellors in rare diseases. These services are available mainly in urban areas. The lack of specialists is one of the reasons why most patients with rare diseases have delayed diagnosis or experience misdiagnosis. However, the initiative from geneticists and specialist teams from HKL to schedule routine visits in a few other states is valuable to reduce the access gap for rare disease patients. The three main public universities have also introduced genetics in their medical curricula, but more needs to be done at the rest of the 30 medical schools in the country. It is vital for Malaysia to have a clearer rare disease national plan and easily accessible rare disease care management that includes comprehensive testing and treatment [36].

The top three rare diseases in Malaysia are Marfan syndrome, Prader-Willi syndrome, and Osteogenesis Imperfecta. The higher number of individuals with these diseases are due to the visible signs and symptoms of these conditions. Other diseases, such as all mucopolysaccharidoses, must undergo some screening and testing procedures. A few of the tests are expensive, and some of them have to be sent abroad, which adds cost and time. Testing for rare diseases requires high infrastructure cost, which includes special equipment or instruments, reagents, and human resource skills or training. In previous years, Malaysia has not had the capacity for in-house testing. The majority of lysosomal disorder testing had to be sent to Australia and Taiwan. However, since 2014, these tests can be carried out at the Institute for Medical Research, which reduces the cost from AUD 2,5000 to only AUD 240–750 (MYR 800–2,000).

## Availability and mobilisation of resources

Funding for orphan drugs remains a massive challenge in implementing UHC. Almost all orphan drugs, especially the enzyme replacement therapy group, are extremely expensive, but this should not be used as an excuse to deny the rights of rare disease patients. In Thailand, *imiglucerase* is listed in the National List of Essential Medicines for the treatment of Gaucher disease type 1, even though the incremental cost-effectiveness ratio (ICER) was 50 times higher (THB 6,300,000 per quality-adjusted life year) than the cost-effectiveness threshold (THB 160,000) [37]. However, in Malaysia, only a small number of patients have access to proper treatment due to the high cost and limited government funding for rare disease treatments. For example, the Malaysian government subsidises only a few enzyme replacement therapies, such as *alglucosidase* and *idursulfase*, as listed in the MOH Medicine Formulary [38]. Moreover, this limited funding is only available for select patients who fulfil the criteria and only through hospitals run by the MOH [39]. Other patients have to queue for the additional fund, and sadly, some of these patients have to wait more than three years to receive their orphan drugs.

## Collaboration with local and other global networks

Rare disease societies in Malaysia play a vital role in minimising the gap in the accessibility of treatment. They play an active role in raising awareness and organising fundraisings. Indeed, there is a need to support easy access to orphan drugs (especially for ERT) for patients who are not in the MOH's list fund. The donation fund is also used to support medical costs, medical equipment, and laboratory tests. Cooperation with the mass media, pharmaceutical industries, and other private agencies is very encouraging, as more Malaysians are aware of rare diseases. Since 2015, Malaysian rare disease advocacy groups have also joined Rainbow Across Borders to act as a regional umbrella alliance for rare diseases. They promote regional collaboration and networking among patient support organisations from the Asia-Pacific region. Aside from empowering patient support, this large organisation also allows members to bond and exchange experiences about their efforts in rare disease patient advocacy groups in Malaysia.

## Limitations of the study

The cross-sectional study design has inherent internal validity for accurately estimating the prevalence of rare disease. However, rare disease in Malaysia is usually treated in a specialised service (genetic clinic). Therefore, this study collected data from public health facilities (MOH and non-MOH hospitals) as only one private facility provided services for rare disease (genetic clinic). Although the prevalence of rare disease is still underestimated, as there are patients who are not yet diagnosed, we believe this is the best estimate for the country.

Public information and statistical data are from sources we deemed to be reliable based on current data and historical trends. Information from communications and interviews is assumed to be reliable. Any such predictions are subject to inherent risks and uncertainties.

## Conclusions

Generally, Malaysia has made significant progress in the management of rare diseases, but there are still opportunities for development in crucial areas. To improve the management of rare diseases, the structure and national plan of the rare disease must be clear and understood by all parties. This development will create better coordination and collaboration between the stakeholders. It is also crucial to have a national rare disease registry, as it could provide the necessary data for further action and foster world-class research on rare diseases. Equally important, efforts must be made to ensure resources such as funding, experts (geneticists) and genetic clinics are made available across the country. Ultimately, if all healthcare providers, governance and politicians work together to manage rare diseases, we will see an improvement of rare disease patient outcomes.

## Supporting information

**S1 Data.**
(XLSX)

**S1 File.**
(DOCX)

## Acknowledgments

We acknowledge the assistance and expertise of the Medical Development Division, Ministry of Health Malaysia, Pharmaceutical Services Division, National Pharmaceutical Regulatory Agency, Malaysian Rare Disorders Society and Malaysia Lysosomal Diseases Association.

## Author Contributions

**Methodology:** Azuwana Supian.

**Resources:** Asrul Akmal Shafie, Mohamed Azmi Ahmad Hassali, Lock-Hock Ngu, Meow-Keong Thong, Hatijah Ayob, Nathorn Chaiyakunapruk.

**Supervision:** Asrul Akmal Shafie, Mohamed Azmi Ahmad Hassali, Nathorn Chaiyakunapruk.

**Validation:** Asrul Akmal Shafie, Lock-Hock Ngu.

**Writing – review & editing:** Asrul Akmal Shafie, Lock-Hock Ngu, Meow-Keong Thong, Nathorn Chaiyakunapruk.

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
