## [Decision Letter · Decision Letter 0]

6 Aug 2019

PONE-D-19-17740

Rare disease in Malaysia: Challenges and solution

PLOS ONE

Dear Dr. SHAFIE,

Thank you for submitting your manuscript to PLOS ONE. After careful consideration, we feel that it has merit but does not fully meet PLOS ONE’s publication criteria as it currently stands. Therefore, we invite you to submit a revised version of the manuscript that addresses the points raised during the review process.

Reviewer 1 recommended rejection, so please pay special attention to the requirements for publication that this reviewer requests. Please conduct a literature search to determine if the results of this paper are 'outdated' as Reviewer 1 suggests. I believe that Reviewer 1's requests that some of the source material, particularly the questionnaire, and the literature supporting the triangulation process are reasonable requirements for publication. I will not of course automatically defer to Reviewer 1's opinion that the interviews be redone, since that would amount to redoing the research and would require a new submission of a new manuscript.

We would appreciate receiving your revised manuscript by Sep 20 2019 11:59PM. To enhance the reproducibility of your results, we recommend that if applicable you deposit your laboratory protocols in protocols.io, where a protocol can be assigned its own identifier (DOI) such that it can be cited independently in the future. For instructions see: http://journals.plos.org/plosone/s/submission-guidelines#loc-laboratory-protocols

We look forward to receiving your revised manuscript.

Kind regards,

James M. Lightwood

Academic Editor

PLOS ONE

Journal Requirements:

2. Please include copies of the interview guide(s) used in the study, in both the original language and English, as Supporting Information, or include a citation if they have been published previously.

3. Please describe in more detail the literature search strategy used in your study (including search terms and dates of the searches), and describe more clearly the results obtained from the literature search. Specifically, please name the sources of information for the data in Table 3.

Reviewers' comments:

Reviewer's Responses to Questions

**Comments to the Author**

1. Is the manuscript technically sound, and do the data support the conclusions?

Reviewer #1: Partly

Reviewer #2: Partly

2. Has the statistical analysis been performed appropriately and rigorously? 

Reviewer #1: N/A

Reviewer #2: N/A

3. Have the authors made all data underlying the findings in their manuscript fully available?

Reviewer #1: No

Reviewer #2: No

4. Is the manuscript presented in an intelligible fashion and written in standard English?

Reviewer #1: No

Reviewer #2: Yes

5. Review Comments to the Author

Reviewer #1: 1. The information is outdated. It needs to be updated by revisiting the literature and redoing the interviews.

2. The checklist for qualitative research studies has not been filled.

3. It is difficult to assess how much data this paper has captured. It appears quite thin. The questionnaire needs to be included as a supplementary file. Also, provide detailed supplementary files of the interviews, as in the recent paper “The role of patient organizations in the rare disease ecosystem in India: An interview-based study.” Orphanet J Rare Dis. 14:117 (2019). Make a list of the literature which was consulted in the triangulation process, and include this as a supplementary file.

4. The English needs improvement. There are many awkward phrases, such as in Line 112, Line 108. Some lines are confusing such as Line 121 -Line 122. Even quotes should be edited for correct English.

Reviewer #2: In the manuscript, Shafie et al. report the current situation of and the recent advances in rare disease realm in Malaysia via cross-sectional study, document-survey and interviews. Authors gathered the information from the multiple points of view, including healthcare system and governance, clinical expertise and patient management, newborn screening, funding and rare disease awareness and patient advocacy, which has high importance in terms of strategic development of new healthcare policies and R&D programs in developing countries, however, the manuscript still holds major and minor issues of concern.

Major Comments:

- The limitation of this cross-sectional study/analysis is its lack of estimation of the numbers of rare diseases/patients expanding toward the entire country, which lead to their underestimation. Rare disease patients could be seen and treated in the other general private and public hospitals than the Ministry of Health Malaysia’s facilities or university hospitals. In order to claim that “they (funding in Malaysia) are not sufficient to cover the high number of rare disease patients”, the estimation would be indispensable.

- The WHO definition of rare diseases has to be referenced. In Europe, some diseases are excluded from the rare disease definition, such as infection and drug hypersensitivities, which handling should be described in the manuscript. And also, authors should mention the diseases which prevalence is less than 0.65% of the population.

- The demographics of the respondents should be shown as well. Especially, the department which the respondents belong to could bias the number of diseases/patients in rare disease realms.

Minor comments:

If space permits, the paper would benefit from the following points of view;

- Comparison of newborn screening tests covered by UHC between Malaysia and the other country which has implemented UHC.

- Research budget in rare diseases research in Malaysia and potential collaboration with International Rare Diseases Research Consortium (IRDiRC).

- Additional description of the potential undiagnosed diseases patients. Authors could refer the undiagnosed disease programs in the other Asian Pacific countries (Western Australia, Japan, etc.).

6. PLOS authors have the option to publish the peer review history of their article (what does this mean?). If published, this will include your full peer review and any attached files.

Reviewer #1: Yes: Dr. Gayatri Saberwal, Dr. Mohua Chakraborty Choudhury

Reviewer #2: No

---

## [Author Response · Author response to Decision Letter 0]

19 Sep 2019

Rare disease in Malaysia: Challenges and solutions

Response to Reviewers

1. Is the manuscript technically sound, and do the data support the conclusions?

Reviewer #1: Partly

Reviewer #2: Partly

Response 

Experiments (methodology) were described in protocols.oi 

DOI: dx.doi.org/10.17504/protocols.io.6zdhf26

https://www.protocols.io/view/rare-disease-in-malaysia-6zdhf26

2. Has the statistical analysis been performed appropriately and rigorously? 

Reviewer #1: N/A

Reviewer #2: N/A

Response: not applicable in this study

3. Have the authors made all data underlying the findings in their manuscript fully available?

Reviewer #1: No

Reviewer #2: No

Response : 

Data Availability Statement: data can be access upon article acceptance 

4. Is the manuscript presented in an intelligible fashion and written in standard English?

Reviewer #1: No

Reviewer #2: Yes

Response : 

We have now send the manuscript to a different professional English proof reading service (American Journal Expert- the certificate as attached) 

Reviewer #1:

1. The information is outdated. It needs to be updated by revisiting the literature and redoing the interviews.

Response : 

The reference is now updated in the revised manuscript. Anothre four new references are added.

2. The checklist for qualitative research studies has not been filled.

Response : 

The reference is now updated in the revised manuscript. Another four new references are added.

3. It is difficult to assess how much data this paper has captured. It appears quite thin. The questionnaire needs to be included as a supplementary file. Also, provide detailed supplementary files of the interviews, as in the recent paper “The role of patient organizations in the rare disease ecosystem in India: An interview-based study.” Orphanet J Rare Dis. 14:117 (2019). Make a list of the literature which was consulted in the triangulation process, and include this as a supplementary file.

Response : 

The semi-structured interview guide is now included in supporting information (attachment)

4. The English needs improvement. There are many awkward phrases, such as in Line 112, Line 108. Some lines are confusing such as Line 121 -Line 122. Even quotes should be edited for correct English.

Response : 

We have now send the manuscript to a different professional English proof reading service (American Journal Expert)

Reviewer #2: 

In the manuscript, Shafie et al. report the current situation of and the recent advances in rare disease realm in Malaysia via cross-sectional study, document-survey and interviews. Authors gathered the information from the multiple points of view, including healthcare system and governance, clinical expertise and patient management, newborn screening, funding and rare disease awareness and patient advocacy, which has high importance in terms of strategic development of new healthcare policies and R&D programs in developing countries, however, the manuscript still holds major and minor issues of concern.

1. The limitation of this cross-sectional study/analysis is its lack of estimation of the numbers of rare diseases/patients expanding toward the entire country, which lead to their underestimation. Rare disease patients could be seen and treated in the other general private and public hospitals than the Ministry of Health Malaysia’s facilities or university hospitals. In order to claim that “they (funding in Malaysia) are not sufficient to cover the high number of rare disease patients”, the estimation would be indispensable.

Response : 

There is currently no epidemiological estimates for rare disease prevalence in Malaysia. Hence, the motivation for the study, we agree that cross sectional study design has inherent internal validity to accurately estmate disease with very rare prevalence. However, rare disease in Malaysia is usually treated in specialized service (genetic clinic).

This study collected data from public health facilities as only one private facility provided the service for rare disease (genetic clinic).

Public health facilities: 

a. The Ministry of Health hospitals

b. Non- Ministry of Health hospitals (teaching hospitals from the Ministry of Education)

Although the prevalent is still underestimated as there are patients who are not yet diagnosed, we believed this is the best estimate for the country. We have now added a new section to our manuscript to discuss this limitation.

2. The WHO definition of rare diseases has to be referenced. In Europe, some diseases are excluded from the rare disease definition, such as infection and drug hypersensitivities, which handling should be described in the manuscript. And also, authors should mention the diseases which prevalence is less than 0.65% of the population.

Response : 

The WHO definition of rare disease already stated in the result:

To date, there is no official definition of rare disease for Malaysia yet. Even though the Malaysian Rare Disorders Society (MRDS), a voluntary organisation defines rare disease as a disease that affects less than 1 per 4000 in the community [19], feedback from the MOH however informally used the WHO definition of rare disease, any disease that has a prevalence between 0.65 - 1% of the population, and the diagnosis and treatment are complicated

3. The demographics of the respondents should be shown as well. Especially, the department which the respondents belong to could bias the number of diseases/patients in rare disease realms.

Response : 

Demographic of respondents added into the manuscript

A total of twenty-six respondents were successfully interviewed in this study. Seven respondents were policymakers (three officers from the Pharmacy Policy & Strategic Planning Division, two officers from the Pharmacy Practice & Development Division and two officers from the National Pharmaceutical Regulatory Agency). Another four respondents were from genetic clinic (three from the Ministry of Health and one from the a university hospital and 12 responders were from the pharmaceutical industry as licensed holders of few orphan drugs. Meanwhile, three responders from the advocacy group were the presidents of leading rare disease society in Malaysia (Malaysian Rare Disorders Society, Malaysia Lysosomal Diseases Association and Malaysia Metabolic Society). However, two respondents from genetic clinics did not respond, and one was on an extended leave.

4. If space permits, the paper would benefit from the following points of view;

Comparison of newborn screening tests covered by UHC between Malaysia and the other country which has implemented UHC.

Response : 

Comparison between Malaysia and the other country which has implemented UHC already discussed in our previous publication:

Shafie, A. A., Chaiyakunapruk, N., Supian, A., Lim, J., Zafra, M., & Hassali, M. A. (2016). State of rare disease management in Southeast Asia. Orphanet journal of rare diseases, 11(1), 107. 

doi:10.1186/s13023-016-0460-9 

https://www.ncbi.nlm.nih.gov/pmc/articles/PMC4969672/

---

## [Decision Letter · Decision Letter 1]

24 Oct 2019

PONE-D-19-17740R1

Rare disease in Malaysia: Challenges and solutions

PLOS ONE

Dear Dr. SHAFIE,

Thank you for submitting your manuscript to PLOS ONE. After careful consideration, we feel that it has merit but does not fully meet PLOS ONE’s publication criteria as it currently stands. Therefore, we invite you to submit a revised version of the manuscript that addresses the points raised during the review process.

We would appreciate receiving your revised manuscript by Dec 08 2019 11:59PM. To enhance the reproducibility of your results, we recommend that if applicable you deposit your laboratory protocols in protocols.io, where a protocol can be assigned its own identifier (DOI) such that it can be cited independently in the future. For instructions see: http://journals.plos.org/plosone/s/submission-guidelines#loc-laboratory-protocols

We look forward to receiving your revised manuscript.

Kind regards,

James M. Lightwood

Academic Editor

PLOS ONE

Additional Editor Comments (if provided):

Reviewer 2 requests a minor revision, which concerns more clarity on the definition used for rare disease in the manuscript, and how that related to the WHO criteria, and some discussion of how the problem of measuring and setting standards for the study of rare disease in Malaysia should be handled, given the problems with reporting. I think this is a valid point and important to address, especially because the authors state in their response to reviewers that the motivation for the study is to initiate adequate research on, and surveillance for, rare diseases in Malaysia.

I do think the revision does display lack of clarity on the issue of the definition of rare diseases in Malaysia. For example, in the Abstract, under Results section, you state in lines 54 to 56:

' Malaysia uses the World Health Organisation’s (WHO) definition of a rare disease as "any disease that is prevalent in 0.65% to 1% of the population and for which and the diagnosis and treatment are complicated”. '

But in the Results section, under Malaysian Healthcare System and Governance, lines 185 to 188, you state:

" To date, there is no official definition of rare disease in Malaysia. Although the Malaysian Rare Disorders Society (MRDS), a voluntary organisation, defines rare disease as a disease that affects less than 1 in 4,000 people in a community [20], feedback from the MOH has informally used the WHO definition of rare disease, which is any disease that is prevalent in 0.65% to 1% of the population, and the diagnosis and treatment are complicated. "

So, even though, apparently, the only effective definition of rare disease in Malaysia is the WHO definition, but this is adopted by a voluntary organization, MRDS. The statement that 'Malaysia uses' in the Abstract could be criticized as being vague or misleading. Is there evidence that the MRDS definition is widely used in Malaysia in the absence of an official definition? In any case, I think the statement in the Abstract needs to be clarified or qualified. I think the issue of the definition of rare disease in Malaysia needs to be more precisely described and Reviewer 2's concerns be addressed.

Reviewers' comments:

Reviewer's Responses to Questions

**Comments to the Author**

1. If the authors have adequately addressed your comments raised in a previous round of review and you feel that this manuscript is now acceptable for publication, you may indicate that here to bypass the “Comments to the Author” section, enter your conflict of interest statement in the “Confidential to Editor” section, and submit your "Accept" recommendation.

Reviewer #1: (No Response)

Reviewer #2: (No Response)

2. Is the manuscript technically sound, and do the data support the conclusions?

Reviewer #1: (No Response)

Reviewer #2: Yes

3. Has the statistical analysis been performed appropriately and rigorously? 

Reviewer #1: (No Response)

Reviewer #2: N/A

4. Have the authors made all data underlying the findings in their manuscript fully available?

Reviewer #1: (No Response)

Reviewer #2: Yes

5. Is the manuscript presented in an intelligible fashion and written in standard English?

Reviewer #1: (No Response)

Reviewer #2: Yes

6. Review Comments to the Author

Reviewer #1: (No Response)

Reviewer #2: I have had a chance to review the revisions made to the manuscript by Shafie et al. entitled “Rare disease in Malaysia: Challenges and solutions”. Overall, the manuscript has improved but I wish to make the following point:

As commented by Ed2, authors must review the “original” source (≠ref #11) of the WHO definition of rare diseases (White paper? Website?). From the original WHO source(s), authors would find the reason why WHO set “0.65-1.00%” for the prevalence of rare disease definition, and should provide considerable explanation about the reason why authors excluded the diseases which prevalence is less than 0.65% of the population from this survey. In order to promote creation of a clear rare disease national plan in Malaysia, solution(s) should be considered not only for the “relatively common” rare disease patients but also for people living with rarer diseases.

7. PLOS authors have the option to publish the peer review history of their article (what does this mean?). If published, this will include your full peer review and any attached files.

Reviewer #1: No

Reviewer #2: No

---

## [Author Response · Author response to Decision Letter 1]

14 Nov 2019

Rare disease in Malaysia: Challenges and solutions

Response to Reviewers 2

Additional editor comments I do think the revision does display lack of clarity on the issue of the definition of rare diseases in Malaysia. For example, in the Abstract, under results section, you state in lines 54 to 56

Response:

Change in the abstract and results

1) Abstract:

Malaysia has no official definition of rare disease yet. However, the government frequently uses the World Health Organisation’s (WHO) definition of a rare disease as "any disease that is prevalent in 0.65% to 1% of the population”. Currently, there are 13 rare disease specialists and a dozen medical doctors in genetic clinics around Malaysia. From the survey, 1,249 patients diagnosed with rare diseases in public hospitals. Only 60% received their medications or supplements, and the rest continued with symptomatic treatment.

2. Results:

To date, there is no official definition of rare disease in Malaysia. Although the Malaysian Rare Disorders Society (MRDS), a voluntary organisation, defines rare disease as a disease that affects less than 1 in 4,000 people in a community [20], the MOH has routinely used the WHO definition of rare disease, which is ‘any disease that is prevalent in 0.65% to 1% of the population, and the diagnosis and treatment are complicated’ in their operation [11]. However, the MOH currently in the process of reviewing the rare disease definition for Malaysia.

Issue 6 

As commented by Ed2, authors must review the “original” source (≠ref #11) of the WHO definition of rare diseases (White paper? Website?). From the original WHO source(s), authors would find the reason why WHO set “0.65-1.00%” for the prevalence of rare disease definition, and should provide a considerable explanation about the reason why authors excluded the diseases which prevalence is less than 0.65% of the population from this survey. In order to promote creation of a clear rare disease national plan in Malaysia, solution(s) should be considered not only for the “relatively common” rare disease patients but also for people living with rarer diseases.

Response:

We agreed that the review of the original source could shed light on the genesis of the definition. In this study, feedback from the Ministry of Health Malaysia indicates that there is no official definition of rare disease in Malaysia yet. However, the ministry provides us with definition that they have informally adopted i.e. “any disease that is prevalent in 0.65% to 1% of the population, and the diagnosis and treatment are complicated”. This is purportedly based on the definition by the World Health Organisation (WHO). The ministry also attached the list of rare diseases together in their feedback. Usually, we do not argue the decision and policy statement from the ministry.

<refer to the attachment, it is not to be published as it is confidential>

Unfortunately, we failed to find the original sources of WHO definition but only a sentence from Song et.al (2012) in their introduction.

Song, P., et al. (2012). "Rare diseases, orphan drugs, and their regulation in Asia: Current status and future perspectives." Intractable Rare Dis Res 1(1): 3-9.

---

## [Editor Report · Decision Letter 2]

11 Dec 2019

PONE-D-19-17740R2

Rare disease in Malaysia: Challenges and solutions

PLOS ONE

Dear Dr. SHAFIE,

Thank you for submitting your manuscript to PLOS ONE. After careful consideration, we feel that it has merit but does not fully meet PLOS ONE’s publication criteria as it currently stands. Therefore, we invite you to submit a revised version of the manuscript that addresses the points raised during the review process.

Reviewer 2 recommends acceptance of the revised manuscript, and I agree that you have adequately addressed Reviewer 2's comments. 

Reviewer 1 has one remaining concern that I think should be addressed, which concerns the World Health Organization's definition of 'rare disease' that seems to implicitly asserted, or perhaps implied, in lines 187 to 192 of the revised manuscript, starting with "feedback from the MOH has informally used the WHO definition of rare disease, which is any disease that is prevalent in 0.65% to 1% of the population, and the diagnosis and treatment are complicated [11]."

Reviewer 1 requests a direct reference to the WHO definition, and an explanation of why diseases with less that 0.65% prevalence are omitted from your analysis. I think this issue needs to be addressed before the manuscript is accepted for publication. 

I read your reference 11, and  it's references that can reasonably be supposed to support the WHO definition stated in your manuscript, but could not find a reference to any WHO document. I searched the WHO site and did not find a section or document on their website that gave any definition. I did find a document that gave a definition adopted by the  Europe Union, which defines a rare disease as one that affects less than 0.1% 'of the population', which presumably refers to its prevalence.  See 'COMMUNICATION FROM THE COMMISSION TO THE EUROPEAN PARLIAMENT, THE COUNCIL, THE EUROPEAN ECONOMIC AND SOCIAL

COMMITTEE AND THE COMMITTEE OF THE REGIONS on Rare Diseases: Europe's challenges'

https://op.europa.eu/en/publication-detail/-/publication/c8a042d8-ffb9-4b01-9c91-c1497a2b3fd7/language-en

I wonder whether this WHO definition is really a 'folklore' definition, that is, one that is not official (perhaps it is given in a WHO working paper, or conference), but taken as such in the rare disease community. 

I suggest that the authors conduct a search to back up the assertion in reference 11, that seems to have no clear backing. If they can find a reference, they should give it instead of, or in addition, to reference 11, and discuss why disease below the lower bound of that definition are omitted from consideration in the manuscript.

However, if the authors cannot find a clear reference to an official WHO source, I think Reviewer 1's concern can be addressed by a discussion of the problem posed by the use of such a 'folklore' definition by the rare disease community, and problems this situation (in my view, rather obviously) poses for planning for rare disease treatment in Malaysia and other countries. I think that finding would also be an interesting result that deserves to be discussed in the Results section.

We would appreciate receiving your revised manuscript by Jan 25 2020 11:59PM. To enhance the reproducibility of your results, we recommend that if applicable you deposit your laboratory protocols in protocols.io, where a protocol can be assigned its own identifier (DOI) such that it can be cited independently in the future. For instructions see: http://journals.plos.org/plosone/s/submission-guidelines#loc-laboratory-protocols

We look forward to receiving your revised manuscript.

Kind regards,

James M. Lightwood

Academic Editor

PLOS ONE

---

## [Author Response · Author response to Decision Letter 2]

15 Jan 2020

Response:

We agreed that reviewing the original source could shed light on the genesis of the definition. We did search various WHO documents and publications when writing the manuscript. However, we cannot find any statement to support feedback from the Ministry of Health Malaysia which defines rare disease "any disease that is prevalent in 0.65% to 1% of the population”. 

We suspect that this could be the case of ‘folklore’ definition whereby an ‘unofficial’ statement is adopted by the community and ministry. This does happens occasionally in the health sector. Usually, we do not argue the decision and policy statement from the ministry especially when is made official by them. However, without strong evidence, the practice could pose a substantial challenge to future planning and discourse.

For the purpose of this manuscript and publication, we agreed to change in the abstract, results and discussion part, and also delete the reference number #11.

Below, the amendments to the manuscript:

Amendment 1

Abstract:

Malaysia has no official definition of rare disease yet, and currently in the process of reviewing the rare disease definition for Malaysia. There are 13 rare disease specialists and a dozen medical doctors in genetic clinics around Malaysia mainly in public health facilities. From the survey, 1,249 patients diagnosed with rare diseases in public hospitals. Only 60% received their medications or supplements, and the rest continued with symptomatic treatment………..

Amendment 2

Results:

To date, there is no official definition of rare disease in Malaysia, even though the most prominent rare disease voluntary organisation called the Malaysian Rare Disorders Society (MRDS) defines rare disease as a disease that affects less than 1 in 4,000 people in a community [20]. In this study, feedback from the Ministry of Health Malaysia indicates that there is no official definition of rare disease in Malaysia yet. However, the ministry provides us with definition that they have informally adopted i.e. “any disease that is prevalent in 0.65% to 1% of the population, and the diagnosis and treatment are complicated” which claimed definition by the World Health Organisation (WHO). Nevertheless, this definition is doubtful as there is no concrete evidence from any of the WHO documents and publications. Currently, the MOH still in the process of reviewing the rare disease definition for Malaysia……………….

Amendment 3

Discussion:

In the management of rare diseases, the definition is fundamental to kick off further actions. Each country has its definition of rare diseases according to its prevalence, and it varies across regions. For example, in the USA, a disease is rare when it affects fewer than 200,000 people [1]. In the European Union, a disease considered a rare disease when it affects less than one in 2,000 people [2]. Whereas in East Asia, Taiwan defines a rare disease if it occurs fewer than one in10,000 population [3]. The majority of the countries stated that the rare disease is between 0.01% to 0.05% of their population. These percentages are far below that provided by the MOH “any disease that is prevalent in 0.65% to 1% of the population”, which claimed adopted from WHO. The definition by MOH is debatable and a high chance a ‘folklore’ definition as there were no official written documents or papers stated this. The definition of rare disease cannot be arbitrary as it could have a wrong impression or a misunderstanding among healthcare stakeholders, especially the patients and caregivers. Nevertheless, the MOH now reviewing the country's rare disease definition to be inline with other reference countries as well as neighboring countries, which will encourage cooperation in matters related to the management of rare diseases in the future.

In 2016, Helen Clark, the former administrator of the United Nations Development Programme stated at the 11th Annual International Conferences on Rare Diseases and Orphan Drugs that, “No country can claim to have achieved UHC if it has not adequately and equitably met the needs of those with rare diseases” [4]. This statement is very relevant to countries that implement total universal health coverage, such as Malaysia. The basic principle of UHC is that all diseases should be treated with readily available and essential treatments. Teerawattananon et al. (2016) suggested four building blocks for setting the effective priority of UHC [5]:

1. Governing structure with precise functions and regulation of institutes and their inter-relation;

2. Capacity building programmes by policymakers, researchers, and other stakeholders, including the general public for a better understanding of health priority;

3. Resource availability and mobilisation to support priority setting; and

4. Collaboration with networks of local, international, and global organisations that aim to support UHC policies.

---

## [Editor Report · Decision Letter 3]

11 Mar 2020

Rare disease in Malaysia: Challenges and solutions

PONE-D-19-17740R3

Dear Dr. Shafie,

We are pleased to inform you that your manuscript has been judged scientifically suitable for publication and will be formally accepted for publication once it complies with all outstanding technical requirements.

With kind regards,

Kwasi Torpey, MD PhD MPH

Academic Editor

PLOS ONE
---

## [Editor Report · Acceptance letter]

18 Mar 2020

PONE-D-19-17740R3 

Rare disease in Malaysia: Challenges and solutions 

Dear Dr. SHAFIE:

I am pleased to inform you that your manuscript has been deemed suitable for publication in PLOS ONE. Congratulations! Your manuscript is now with our production department. 

With kind regards,

on behalf of

Professor Kwasi Torpey 

Academic Editor

PLOS ONE